# Frequency-Domain Dynamic Pruning for Convolutional Neural Networks

**Zhenhua Liu[1], Jizheng Xu[2], Xiulian Peng[2], Ruiqin Xiong[1]**
[1]Institute of Digital Media, School of Electronic Engineering and Computer Science, Peking University
[2]Microsoft Research Asia
liu-zh@pku.edu.cn, jzxu@microsoft.com, xipe@microsoft.com, rqxiong@pku.edu.cn

## Abstract

Deep convolutional neural networks have demonstrated their powerfulness in a variety of applications. However, the storage and computational requirements have largely restricted their further extensions on mobile devices. Recently, pruning of unimportant parameters has been used for both network compression and acceleration. Considering that there are spatial redundancy within most filters in a CNN, we propose a frequency-domain dynamic pruning scheme to exploit the spatial correlations. The frequency-domain coefficients are pruned dynamically in each iteration and different frequency bands are pruned discriminatively, given their different importance on accuracy. Experimental results demonstrate that the proposed scheme can outperform previous spatial-domain counterparts by a large margin. Specifically, it can achieve a compression ratio of $8.4\times$ and a theoretical inference speed-up of $9.2\times$ for ResNet-110, while the accuracy is even better than the reference model on CIFAR-10.

## 1 Introduction

In recent years, convolutional neural networks have been performing well in a variety of artificial tasks including image classification, face recognition, natural language processing and speech recognition. Since convolutional neural networks tend to be deeper and deeper which means more storage requirements and floating-point operations, there are many works devoting to simplify and accelerate the deep neural networks.

Some works performed structural sparsity approximation which alter the large sub-networks or layers into shallow ones. Jaderberg et al. [1] proposed to construct a low rank basis of filters by exploiting cross-channel or filter redundancy. [2] took the nonlinear units into account and minimized the reconstruction error of nonlinear responses, subjecting to a low-rank constraint. [3] and [4] employed tensor-decomposition and tucker-decomposition to simplify convolutional neural networks, respectively.

Since operations in high precision are much more time-consuming than those with fewer fix-point values, Courbariaux et al. [5] proposed to constrain activations to $+1$ and $-1$. [6] proposed XNOR-Networks which computed the scaling factor applying to both binary weights and binary input. [7] proposed HORQ Network which recursively computed the quantized residual to reduce the information loss. Methods in [8–11] employed ternary or fixed-point values to compress and accelerate the convolutional neural networks.

Some researchers also employed quantization to reduce the computation of CNNs. [12] utilized k-means clusting to identify the shared weights and limited all the weights that fell into the same cluster sharing the same weight. [13] employed product quantization to implement the efficient inner product computation. [14] extended the quantization method into frequency domain and used a hash function to randomly group frequency parameters into hash buckets and all parameters assigned to

the same hash bucket shared a single value learned with standard backpropagation. [15] decomposed the representations of convolutional filters in frequency domain as common parts (i.e. cluster centers) shared by other similar filters and their individual private parts (i.e. individual residuals). [16] revived a principled regularization method based on soft weight-sharing.

Besides decomposition and quantization, network pruning is also a widely studied and efficient approach. By pruning the near-zero connections and retraining the pruned network, both the network storage and computation can be reduced. Han et al. [17] showed that network pruning can compress AlexNet and VGG-16 by $9\times$ and $13\times$, respectively, with negligible accuracy loss on ImageNet. [18] proposed a dynamic network surgery (DNS) method to reduce the network complexity. Compared with the pruning methods which accomplished this task in a greedy way, they incorporated connection splicing into the surgery to avoid incorrect pruning and made it as a continual network maintenance. They compressed the parameters in LeNet-5 and AlexNet by $108\times$ and $17.7\times$ respectively. To further accelerate the deep convolutional nerural networks, [19] proposed to conduct channel pruning by a LASSO regression based channel selection and least square reconstruction. [16] and [20] process kernel weights in spatial domain and achieve both pruning and quantization in one training procedure. [21], [22] and [23] prune nodes or filters by employing Bayesian point of view and $L_0$ norm regularization.

The pruning methods mentioned above are all conducted in spatial domain. Actually, due to the local smoothness of images, most filters in a CNN tend to be smooth, i.e. there are spatial redundancies. In this paper, we try to fully exploit this spatial correlation and propose a frequency-domain network pruning approach. First we show that a convolution or an inner product can be implemented by a DCT-domain multiplication. Further we apply a dynamic pruning to the DCT coefficients of network filters, since the dynamic method achieves a pretty good performance among the spatial pruning approaches. What's more, due to variant importance of different frequency bands, we compressed them with different rates. Experimental results show that the proposed scheme can outperform previous spatial-domain counterparts by a large margin on several datasets, without or with negligible accuracy loss. Specifically, the proposed algorithm can acquire accuracy gain for the ResNet on CIFAR-10, while achieving an impressive compression and acceleration of the network.

The rest of this paper is organized as follows. In Section 2, we introduce the proposed band-adaptive frequency-domain dynamic pruning scheme. The theoretical analysis of computational complexity is presented in Section 3. Section 4 shows the experimental results on several datasets. Section 5 concludes the paper.

## 2 Frequency-Domain Network Pruning

In this section, we first show how a spatial-domain convolution or inner product can be implemented by multiplication in frequency domain. Here we use 2-D DCT for spatial redundancy removal. Then the proposed frequency-domain dynamic pruning method is introduced. Further, the band-adaptive rate allocation strategy is explained, which prunes different frequency bands discriminatively according to their importances.

### 2.1 Frequency-Domain CNN

We first consider a convolutional layer of a CNN with the input tensor $\mathcal{I} \in \mathbb{R}^{c_{in} \times w_{in} \times h_{in}}$ and convolutional filters $\mathcal{W} \in \mathbb{R}^{c_{in} \times d \times d \times c_{out}}$. For the weight tensor $\mathcal{W}$, the spatial support of each kernel filter is $d \times d$. There are $c_{in}$ input channels and $c_{out}$ output feature maps. We unfold each $c_{in}$ filters into a 1-D vector with a size of $(c_{in} \times d \times d) \times 1$. Then the weight tensor $\mathcal{W}$ is reshaped to a $(c_{in} \times d \times d) \times c_{out}$ matrix $W$ (see Fig. 1).

Let $\mathcal{O} \in \mathbb{R}^{c_{out} \times w_{out} \times h_{out}}$ denote the output of the convolutional layer $\langle \mathcal{I}, \mathcal{W}, * \rangle$, where $w_{out} = \lfloor (w_{in} + 2p - d)/s \rfloor + 1$ and $h_{out} = \lfloor (h_{in} + 2p - d)/s \rfloor + 1$. $p$ and $s$ are the padding and stride parameters, respectively. The input tensor $\mathcal{I}$ can be reshaped into a $(w_{out} \times h_{out}) \times (c_{in} \times d \times d)$ matrix $I$, where each row is the unfolded version of a sub-tensor in $\mathcal{I}$ with the same size of a group of $c_{in}$ filters. Then the convolution can be implemented by a matrix multiplication between $I$ and $W$. The output is given by $O = I \cdot W$, where $O$ has a shape of $(w_{out} \times h_{out}) \times (c_{out})$. The matrix $O$ can be reshaped to the output tensor $\mathcal{O} \in \mathbb{R}^{c_{out} \times w_{out} \times h_{out}}$ by folding each column into a $w_{out} \times h_{out}$ feature map.

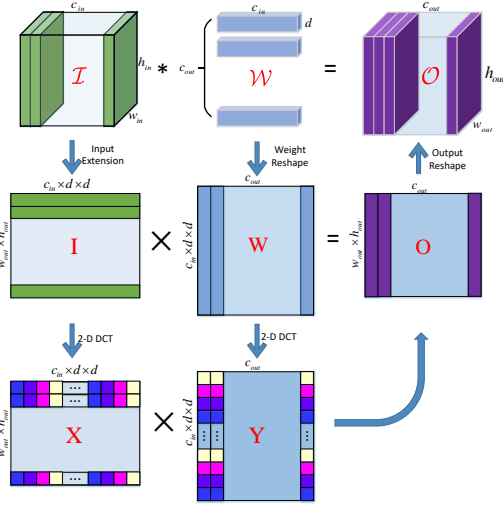

Figure 1: This figure shows the process of convolution in frequency domain.

Suppose $\mathbf{w} = \left[\mathbf{w}_1^T, \mathbf{w}_2^T, \cdots, \mathbf{w}_{c_{in}}^T\right]^T$ is a column vector in $W$, where $\mathbf{w}_k(k \in [1, c_{in}])$ represents the 1-D form of a kernel filter. Then the 2-D DCT whose transformation size is $d \times d$ can be applied to each kernel filter. The operation can be seen as a matrix multiplication $\mathbf{y}_k = B \cdot \mathbf{w}_k$, where $B$ is the Kronecker tensor product of 2-D DCT transformation matrix and itself. The shape of $B$ is $(d \times d) \times (d \times d)$. After obtaining each sub-vector of coefficients $\mathbf{y}_k$, the coefficients matrix $Y$ is generated by reconnecting the sub-vectors and regrouping the vectors $\mathbf{y}$. The input coefficients matrix $I$ can be obtained using the same scheme except that we apply 2-D DCT to the row sub-vectors of $I$. The shapes of $X$ and $Y$ are the same as $I$ and $W$, respectively.

In fact, the 2-D DCT transform of $W$ can be viewed as a matrix multiplication, i.e. $Y = A \cdot W$. $A$ is the Kronecker tensor product of $B$ and a unit matrix $E$ whose size is $c_{in} \times c_{in}$.

$$A = \begin{pmatrix} B & 0 & \ldots & 0 \\ 0 & B & \ldots & 0 \\ \vdots & \vdots & \ddots & \vdots \\ 0 & 0 & \cdots & B \end{pmatrix} \tag{1}$$

On the other hand, we can derive $X^T = A \cdot I^T$. The computations of $X_{i,j}^T$ and $Y_{i,j}$ are shown in Eq. 2 and 3, respectively.

$$X_{i,j}^T = \sum_{k=1}^{c_{in}} \sum_{\ell=1}^{d^2} A_{i,(k-1)\cdot d^2+\ell} I_{(k-1)\cdot d^2+\ell,j}^T \tag{2}$$

$$Y_{i,j} = \sum_{k=1}^{c_{in}} \sum_{\ell=1}^{d^2} A_{i,(k-1)\cdot d^2+\ell} W_{(k-1)\cdot d^2+\ell,j} \tag{3}$$

Since the basis of a 2-D DCT are orthonormal, we can easily derive that both $B$ and $A$ are also orthonormal matrices. As shown in Figure 1, the output matrix $O$ can be computed by directly multiplying $X$ and $Y$, the proof of which is given as follows.

$$\begin{aligned} X \cdot Y &= (A \cdot I^T)^T \cdot (A \cdot W) \\ &= I \cdot A^T \cdot A \cdot W \\ &= I \cdot (A^T \cdot A) \cdot W \\ &= I \cdot W = O \end{aligned}$$

In this way, the convolution in spatial domain is realized by the matrix multiplication in frequency domain.

As for the fully-connected layers, the weights can be viewed as a matrix shape of $c_{in} \times c_{out}$ and the input is a vector shape of $1 \times c_{in}$. The same scheme of convolutional layers can be directly applied to implement an inner product in frequency domain. For those fully-connected layers whose inputs are the outputs of convolutional layers, the 2-D DCT size can be set as the size of feature map in the previous connected convolutional layer. As for other fully-connected layers, the 2-D DCT size can be decided according to the size of input vector. In this paper, we do not apply transformation to the latter kind of fully-connected layers, since the correlations among their weights are not so strong.

## 2.2 Frequency-Domain Dynamic Network Pruning (FDNP)

As mentioned in Section 2.1, we can obtain the transform coefficients of the input $X$ and the weight filters $Y$ in a matrix form. In this section, we show how the proposed frequency-domain dynamic network pruning approach works. In spatial domain, the filters in a CNN are mostly smooth due to the local pixel smoothness in natural images. In frequency domain, this leads to components with large magnitudes in the low frequency bands and small magnitudes in the high frequency bands. The coefficients in frequency domain are more sparse than that in spatial domain, so they have more potential to be pruned while retaining the same crucial information.

In order to represent a sparse model with part of its parameters pruned away, we utilize a mask matrix $T$ whose values are binary to indicate the states of parameters, i.e., whether they are currently pruned or not. Then the optimization problem can be described as

$$\min_{Y,T} \mathcal{L}(\mathcal{D}^{-1}(Y \otimes T)) \quad \text{s.t.} T_{i,j} = f(Y_{i,j}), \tag{4}$$

where $\mathcal{L}(\cdot)$ is the loss function. $\mathcal{D}^{-1}$ denotes the inverse 2-D DCT. $\otimes$ represents the Hadamard product operator and $f(\cdot)$ is the discriminative function which satisfies $f(Y_{i,j}) = 1$ if coefficient $Y_{i,j}$ seems to be important in the current layer and 0 otherwise. Following the dynamic method in [18], we set two thresholds $a$ and $b$ to decide the mask values of coefficients for each layer. And to properly evaluate the importance of each coefficient, an absolute value is utilized. Using function $f(\cdot)$ as below, the coefficients are not pruned forever and have a chance to return during the training process.

$$f(Y_{i,j}^{t+1}) = \begin{cases} 0 & if \ a > |Y_{i,j}^{t+1}| \\ T_{(i,j)}^t & if \ a \le |Y_{i,j}^{t+1}| \le b \\ 1 & if \ b < |Y_{i,j}^{t+1}| \end{cases} \tag{5}$$

The $a$ and $b$ are set according to the distribution of coefficients in each layer, i.e.

$$a = 0.9 * (\mu + \gamma * \sigma) \tag{6}$$
$$b = 1.1 * (\mu + \gamma * \sigma) \tag{7}$$

in which $\mu$ and $\sigma$ are the mean value and the standard deviation of all coefficients in one layer, respectively. $\gamma$ denotes the compression rate of each layer, by which the number of remaining coefficients in each layer is determined.

## 2.3 Band Adaptive Frequency-Domain Dynamic Network Pruning (BA-FDNP)

As components of different frequencies tend to be of different magnitudes, their importances vary for the spatial structure of a filter. Thus we allow varying compression rates for different frequencies. The frequencies are partitioned into $2d - 1$ regions after analyzing the distribution of the coefficients, where $d$ is the transformation size. The compression rate $\gamma_k$ is set for the $k^{th}$ frequency region where $k = u + v$ is the index of a frequency region. A smaller $\gamma_k$ introduces lower threshold and less parameters will be pruned. Since lower frequency components seem to be of higher importance, we commonly assign lower $\gamma_k$ to low-frequency regions with small indices $(u, v)$. Correspondingly, the high frequencies with large indices $(u, v)$, have magnitudes near zero. Larger compression rates will fit them better.

We set the compression rate of $k^{th}$ frequency region $\gamma_k$ with a parameterized function, i.e. $\gamma_k = g(\cdot)$. We adopt the beta distribution in the experiment:

$$g(x; \lambda, \omega) = x^{\lambda-1}(1-x)^{\omega-1} \tag{8}$$

where $x = (k+1)/2d, k \in [0, 2d-1]$.

As we mentioned before, we expect a smaller compression rate for low-frequency components due to the higher importance. In the experiment, we modify function $g(\cdot)$ to be positively related with $x$ by adjusting the values of $\lambda$ and $\omega$.

## 2.4 Training Pruning Networks in Frequency Domain

The key point of training pruning networks in frequency domain is the updating scheme of weight coefficients matrix $Y$. Suppose $\mathbf{w}$ is a kernel filter in $\mathcal{W}$, and $\mathbf{y}$ denotes the corresponding coefficients after 2-D DCT transformation whose shape is $d \times d$. Since 2-D DCT is a linear transformation, the gradient in frequency domain is merely the 2-D DCT transformation of the gradient in spatial domain. The proof is shown in the supplementary material.

$$\frac{\partial \mathcal{L}}{\partial \mathbf{y}} = \mathcal{D}(\frac{\partial \mathcal{L}}{\partial \mathbf{w}}) \tag{9}$$

where $\mathcal{L}$ is the total loss of the network. Then inspired by the method of Lagrange Multipliers and gradient descent, we can obtain a straightforward updating procedure of the filter parameters $Y$ in frequency domain.

$$Y_{(u,v)} \leftarrow Y_{(u,v)} - \beta \frac{\partial \mathcal{L}(Y \otimes T)}{\partial (Y_{(u,v)} T_{(u,v)})} \tag{10}$$

in which $\beta$ is a positive learning rate. To enable the returning of improperly pruned parameters, we update not only the non-zero coefficients, but also the ones corresponding to zero entries of $T$.

The procedure of training an $L-$layers BA-FDNP network can be divided into three phases: feed-forward, back-propagation and coefficient-update. In the feed-forward phase, the input and weight filters are transformed into frequency domain to complete the computation as shown in Section 2.1. During back-propagation, after computing the standard gradient $\frac{\partial \mathcal{L}}{\partial W}$ in spatial domain, 2-D DCT is directly used to obtain the gradient $\frac{\partial \mathcal{L}}{\partial Y}$ in frequency domain. Then we apply dynamic pruning method after updating the coefficients. It should be noticed that we update not only the remained coefficients, but also the ones considered to be unimportant temporarily. So we can give a chance for those improperly pruned parameters to be returned. Repeat these steps iteratively, one can train the BA-FDNP CNN in frequency domain. The process of the proposed algorithm is detailed in the supplementary material.

## 3 Computational Complexity

Given a convolutional layer with $\mathcal{W} \in \mathbb{R}^{c_{in} \times d \times d \times c_{out}}$ as the weight tensor and $\mathcal{Y}$ denotes the compressed coefficients in this layer. Suppose $\eta$ is the ratio of non-zero elements in $\mathcal{Y}$, the number of multiplications in convolution operations is $\eta c_{in} d^2 c_{out} w_{out} h_{out}$ in frequency domain. And each sub-input feature map with a size of $d \times d$ costs $2d \cdot d^2$ multiplications due to the separable 2-D DCT transform. The additional computational cost of 2-D DCT in one layer is $2d \cdot d^2 c_{in} w_{out} h_{out}$ (see Fig. 1). Compared to the original CNN, the theoretical inference speed-up of the proposed scheme is

$$r_s = \frac{c_{in} d^2 c_{out} w_{out} h_{out}}{2d \cdot d^2 c_{in} w_{out} h_{out} + \eta c_{in} d^2 c_{out} w_{out} h_{out}} = \frac{c_{out}}{2d + \eta c_{out}} \tag{11}$$

Suppose $\xi$ is the ratio of non-zero elements in the compressed weights of spatial-domain pruning method. The inference speed-up of the spatial-domain pruning method is

$$r_s{}' = \frac{c_{in} d^2 c_{out} w_{out} h_{out}}{\xi c_{in} d^2 c_{out} w_{out} h_{out}} = \frac{1}{\xi} \tag{12}$$

Eq.11 and 12 give the inference speed-up of one layer, the inference speed-up of whole network is also related to the size of output feature map in each layer besides the compression rates of each layer. Although compared to the spatial-domain pruning method, our scheme has an additional computation cost of transformation, a larger compression ratio can be acquired, i.e. less non-zero elements left in compressed parameters, due to the sparser representation of the weight filters. We will show the detailed results in Section 4.

Table 1: Compression results comparison of our methods with [17] and [18] on LeNet-5.

| LeNet-5 | Parameters | Top-1 Accuracy | Iterations | Compression |
|---|---|---|---|---|
| **Reference** | 431K | 99.07% | 10K | 1× |
| **Pruned [17]** | 34.5K | 99.08% | 10K | 12.5× |
| **Pruned [18]** | 4.0K | 99.09% | 16K | 108× |
| **FDNP (ours)** | 3.3K | 99.07% | 20K | **130×** |
| **BA-FDNP (ours)** | 2.8K | 99.08% | 20K | **150×** |

Table 2: Comparison of the percentage of remaining parameters in each layer after applying [17], [18] and our methods on LeNet-5.

| Layer | Params. | Params.%[17] | Params.%[18] | Params.%(FDNP) | Params.%(BA-FDNP) |
|---|---|---|---|---|---|
| conv1 | 0.5K | 66% | 14.2% | 12.4% | 12% |
| conv2 | 25K | 12% | 3.1% | 2.5% | 2.1% |
| fc1 | 400K | 8% | 0.7% | 0.6% | 0.5% |
| fc2 | 5K | 19% | 4.3% | 3.5% | 3.6% |
| Total | 431K | 8% | 0.9% | **0.77%** | **0.67%** |

## 4 Experimental Results

In this section, we conduct comprehensive experiments on three benchmark datasets: MNIST, ImageNet and CIFAR-10. LeNet, AlexNet and ResNet are tested on these three datasets respectively. We mainly compare our schemes with [17] and [18], which are spatial-domain pruning methods. The compared results of LeNet and AlexNet are directly acquired from the paper and we train the compressed model of ResNet ourselves as the paper introduced since they didn't report the results of ResNet.

The training processes are all performed with the Caffe framework [24]. A pre-trained model is obtained before applying pruning and the learning policy of fine-tuning is the same as the ones while obtaining the pre-trained model if not mentioned specifically. The momentum and weight decay are set to 0.9 and 0.0001 in all experiments. Since LeNet and AlexNet have only a few layers and the kernel size of each layer is different, we set the compression rate $\gamma$ of each layer manually. On the other hand, we set the same $\gamma$ for every layer in ResNet. In the BA-FDNP scheme, the hyperparameters $\lambda$ and $\omega$ are set to 1.0 and 0.8 respectively.

### 4.1 LeNet-5 on MNIST

We firstly apply our scheme on MNIST with LeNet-5. MNIST is a benchmark image classification dataset of handwritten digits from 0 to 9 and LeNet-5 is a conventional neural network which consists of four learnable layers, including two convolutional layers and two fully-connected layers. It is designed by LeCun et al. [25] for document recognition and has $431K$ learnable parameters. The learning rate is set to 0.1 initially and reduced by 10 times for every 4K iterations during training. We use "*xavier*" initialization method and train a reference model whose top-1 accuracy is $99.07\%$ with $10K$ iterations.

While compressing LeNet-5 with FDNP and BA-FDNP, the batch size is set to 64 and the maximal number of iterations is properly increased to $15K$. The comparision of our proposed schemes with [17] and [18] are shown in Table 1. The network parameters of LeNet-5 are reduced by a factor of $130\times$ and $150\times$ with FDNP and BA-FDNP repetively which are much better than [17] and [18], while the classification accuracies are as good.

To better demonstrate the advantage of our schemes, we make layer-by-layer comparisons among [17], [18] and our schemes in Table 2. There is a considerable improvement in every layer due to the transformation. And we can see that the performance can benefit from different compression rates for different frequency bands.

Table 3: Compression results comparison of our methods with [17] and [18] on AlexNet.

| AlexNet | Top-1 Acc. | Top-5 Acc. | Parameters | Iterations | Compression |
|---|---|---|---|---|---|
| **Reference** | 56.58% | 79.88% | 61M | 45K | 1× |
| **Pruned [17]** | 57.23% | 80.33% | 6.8M | 480K | 9× |
| **Pruned [18]** | 56.91% | 80.01% | 3.45M | 70K | 17.7× |
| **FDNP(ours)** | 56.84% | 80.02% | 2.9M | 70K | **20.9×** |
| **BA-FDNP(ours)** | 56.82% | 79.96% | 2.7M | 70K | **22.6×** |

Table 4: Comparison of the percentage of remaining parameters in each layer after applying [17], [18] and our methods for AlexNet.

| Layer | Params. | Params.%[17] | Params.%[18] | Params.%(FDNP) | Paras.%(BA-FDNP) |
|---|---|---|---|---|---|
| conv1 | 35K | 84% | 53.8% | 40.7% | 42.3% |
| conv2 | 307K | 38% | 40.6% | 35.1% | 34.6% |
| conv3 | 885K | 35% | 29.0% | 28.6% | 24.6% |
| conv4 | 664K | 37% | 32.3% | 29.9% | 27.7% |
| conv5 | 443K | 37% | 32.5% | 26.7% | 23.8% |
| fc1 | 38M | 9% | 3.7% | 3.4% | 3.0% |
| fc2 | 17M | 9% | 6.6% | 4.7% | 4.8% |
| fc3 | 4M | 25% | 4.6% | 3.9% | 3.8% |
| Total | 61M | 11% | 5.7% | **4.8** | **4.4%** |

## 4.2 AlexNet on ImageNet

We further examine the performance of our scheme on the ILSVRC-2012 dateset, which has 1.2 million training images and 50K validation images. AlexNet is adopted as the inference network. AlexNet has five convolutional layers and three fully-connected layers. After $450K$ iterations of training, a reference model with 61 million well-learned parameters is generated. While performing FDNP and BA-FDNP, the convolutional layers and fully-connected layers are pruned separately which is also applied in [18]. We run $350K$ iterations for convolutional layers and $350K$ iterations for fully-connected layers respectively. When the weight coefficients of convolutional layers are pruning, the weight coefficients of the fully-connected layers update as well but not be pruned and vice versa. We use a learning rate of 0.01 and reduced it by 10 times for every 100K iterations. The batch size is set to 32 and we use "*gaussian*" initialization method for the training.

As we mentioned in section 2.1, FDNP and BA-FDNP are appplied to the convolutional layers as well as the first fully-connected layer whose input is the output feature map of convolutional layer and the other fully-connected layers are pruned in spatial domain using the method in [18]. Table 3 shows the comparison of our schemes with [17] and [18] on AlexNet. Our FDNP and BA-FDNP methods achieve $20.9\times$ and $22.6\times$ compression ratios which are better than the spatial-domain pruning methods. Besides, the classification accuracies of our compressed schemes are still comparable with the compared methods and better than the reference model.

We compare the percentage of remaining parameters in each layer of AlexNet after applying [17], [18] and our methods in Table 4. Our methods pruned more parameters on every single layer. Although we utilize the method in [18] among the last two fully-connected layers, our compressed model achieves the lager compression ratios on these two layers due to the more expressive capacity of the other layers.

## 4.3 ResNet on CIFAR-10

To further demonstrate the effectiveness of our scheme, we apply it to the modern neural network ResNet [26] on CIFAR-10. CIFAR-10 is also a classification benchmark dataset containing a training set of $50K$ images and a test set of $10K$ images. During training, we use the same data augmentation like in [27], which contains flip and translation. ResNet-20 and ResNet-110 are conducted in this

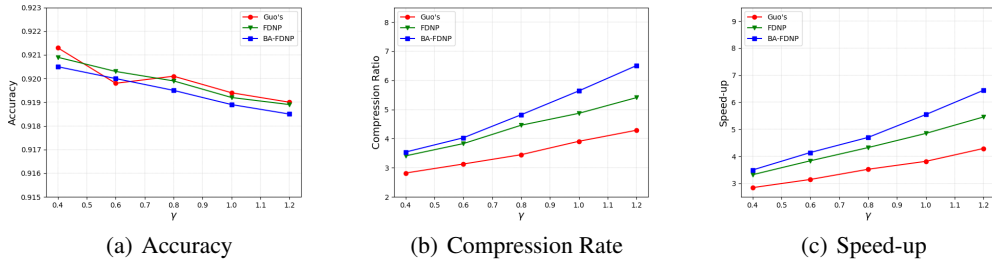

(a) Accuracy        (b) Compression Rate        (c) Speed-up

Figure 2: This figure shows the accuracies, compression rates and theoretical inference speed-up of ResNet-20 under different $\gamma$.

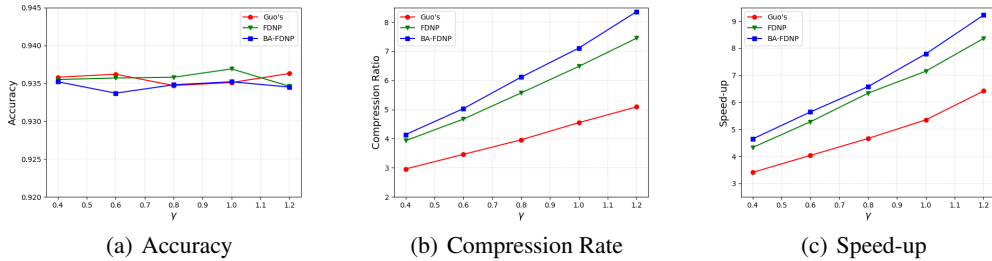

(a) Accuracy        (b) Compression Rate        (c) Speed-up

Figure 3: This figure shows the accuracies, compression rates and theoretical inference speed-up of ResNet-110 under different $\gamma$.

experiment. The learning rate is 0.1 and reduced by 10 times for every 40K iterations. The "msra" initialization method is adopted in this experiment. After $100K$ iterations of training, the top-1 accuracies of two reference models are $91.86\%$ and $93.41\%$ respectively.

We apply the method in [18], FDNP and BA-FDNP to the reference models of ResNet-20 and ResNet-110 separately. The batch size is set to 100 and the maximal number of iterations is set to $150K$. While applying FDNP and BA-FDNP, we employ the spatial-domain dynamic pruning method in [18] to compress the convolutional layers whose kernel sizes are $1 \times 1$.

Fig.2 and 3 show the performances of [18] and our proposed schemes under different $\gamma$. It can be seen that a larger $\gamma$ prominently improves the compression ratio, while coming at a cost of a little decreased accuracy. Under the same condition, FDNP achieve larger compression ratios and inferece speed-up rates while the accuracies are nearly the same as or even better than [18]. When we compress the model using different compression rates for different frequency bands, the performance can be better. While keeping the accuracies the same as the reference models, our BA-FDNP scheme can compress ResNet-20 and ResNet-110 by a factor of $6.5\times$ and $8.4\times$ respectively. In the meantime, the theoretical inference speed-up ratios are $6.4\times$ and $9.2\times$ respectively, which means both the storage requirements and the FLOPs can be well reduced. The interesting point is that the speed-up ratio of ResNet-110 is ever larger than the compression ratio even with the additional computational cost of 2-D DCT transformation. We consider that it owe to our schemes pruning more coefficients of layers who own larger size of output feature map.

Figure.4(a) shows the number of pruned and remaining parameters of each layer in ResNet-20 when $\gamma$ is set to 1.2. The proportions of the remaining parameters in different layers are different though we set the same compression rate. The energy histogram of the coefficients before and after BA-FDNP is shown in Figure.4(b). The energy of coefficients is more concentrated on the lower frequency bands after pruning as we pruned more higher frequency coefficients. By this result, it appears that the lower frequency components tend to be of higher importance.

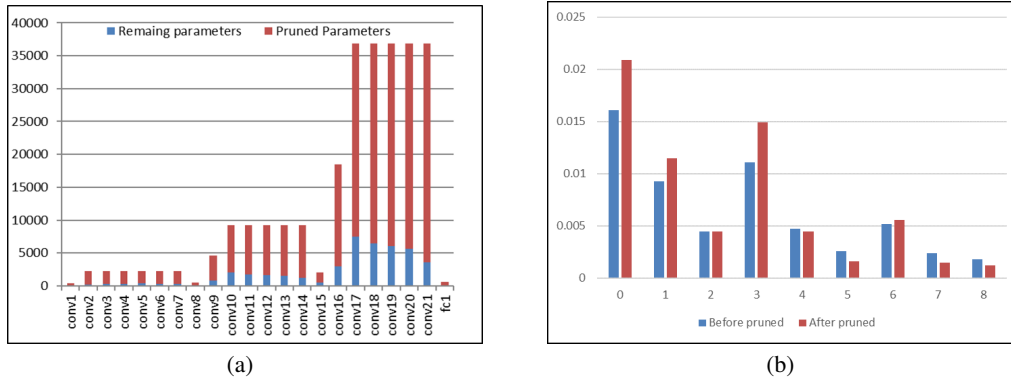

Figure 4: This figure shows (a) the number of pruned and remaining parameters of each layer in ResNet-20 after applying BA-FDNP, (b) the energy histogram of coefficients in each band before and after BA-FDNP in ResNet-20.

## 5 Conclusion

In this paper, we propose a novel approach to compress the convolutional neural networks by dynamically pruning the unimportant weight coefficients in frequency domain. We firstly give an implementation of CNN in frequency domain. The coefficients can be efficiently pruned since they are sparser after 2-D DCT transformation and many spatial-domain pruning methods can be applied. What's more, we set different compression rates for different frequency bands due to the variant importance. Our BA-FDNP scheme achieves a $8.4\times$ of compression and a $9.2\times$ of acceleration for ResNet-110 respectively without any loss of accuracy, which outperforms the previous pruning methods by a considerable margins. In the future, we will consider to exploit the correlations among different channels and employ 3-D transform to further compress and accelerate the convolutional neural networks. Besides, the quantization and Huffman-coding can also be applied to the coefficients in frequency domain.

**Acknowledgemetns** This work was part supported by the National Key Research and Development Program of China (2017YFB1002203), the National Natural Science Foundation of China (61772041), the Beijing Natural Science Foundation (4172027), and also by the Cooperative Medianet Innovation Center. This work was done when Z. Liu was with Microsoft Research Asia.

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
