[Supplementary Material]

# Frequency-Domain Dynamic Pruning for Convolutional Neural Networks(Supplementary Material)

**Zhenhua Liu**[1], **Jizheng Xu**[2], **Xiulian Peng**[2], **Ruiqin Xiong**[1]

[1]Institute of Digital Media, School of Electronic Engineering and Computer Science, Peking University
[2]Microsoft Research Asia

`liu-zh@pku.edu.cn, jzxu@microsoft.com, xipe@microsoft.com, rqxiong@pku.edu.cn`

In this document we give the proof of that the gradient in the frequency domain is merely the transformation of the gradient in spatial domain. We also detail the algorithm of the proposed compression scheme in Section 2.4.

## 1 Proofs of the Inference

Suppose $\mathbf{w}$ is one kernel filter of convolutional neural network and $\mathbf{y}$ is the corresponding transform coefficients. The 2-D DCT transformation is defined as:

$$\mathbf{y}_{u,v} = c(u)c(v)\sum_{i=0}^{d-1}\sum_{j=0}^{d-1}\mathbf{w}_{i,j}s(i,j,u,v) \tag{1}$$

where

$$s(i,j,u,v) = \cos[\frac{(2i+1)\pi}{2d}u]\cos[\frac{(2j+1)\pi}{2d}v] \tag{2}$$

$$c(u) = \begin{cases} \sqrt{\frac{1}{d}} & if\ u = 0 \\ \sqrt{\frac{2}{d}} & if\ u \neq 0 \end{cases} \tag{3}$$

The inverse 2-D DCT converts $\mathbf{y}$ from the frequency domain back to the spatial domain, which can be described as bellow.

$$\mathbf{w}_{i,j} = \sum_{u=0}^{d-1}\sum_{v=0}^{d-1}c(u)c(v)\mathbf{y}_{u,v}s(i,j,u,v) \tag{4}$$

Following Equation **??**, we express the gradient of parameters in the spatial domain with respect to their counterparts in the frequency domain.

$$\frac{\partial \mathbf{w}_{i,j}}{\partial \mathbf{y}_{u,v}} = c(u)c(v)s(i,j,u,v) \tag{5}$$

Using standard back-propagation, we can derive the gradient with respect to the filter parameters in the spatial domain, $\frac{\partial \mathcal{L}}{\partial \mathbf{w}_{i,j}}$. By the chain rule with Equation **??**, we infer the gradient of $\mathcal{L}$ in the frequency domain:

$$\frac{\partial \mathcal{L}}{\partial \mathbf{y}_{u,v}} = \sum_{i=0}^{d-1}\sum_{j=0}^{d-1}\frac{\partial \mathcal{L}}{\partial \mathbf{w}_{i,j}}\frac{\partial \mathbf{w}_{i,j}}{\partial \mathbf{y}_{u,v}}$$

$$= c(u)c(v)\sum_{i=0}^{d-1}\sum_{j=0}^{d-1}\frac{\partial \mathcal{L}}{\partial \mathbf{w}_{i,j}}s(i,j,u,v)$$

Comparing with Equation **??**, we see that the gradient in frequency domain is merely the 2-D DCT transformation of the gradient in spatial domain.

$$\frac{\partial \mathcal{L}}{\partial \mathbf{y}} = \mathcal{D}(\frac{\partial \mathcal{L}}{\partial \mathbf{w}}) \tag{6}$$

## 2 Algorithm of the Proposed Scheme

In Section 2.4, we has proposed an algorithm for compressing CNN, which dynamic prunes the coefficients in frequency domain. Alg.**??** summarizes the training procedures of the proposed scheme.

---

**Algorithm 1** Training an L-layers CNN with BA-FDNP

---

**input** training data $\mathcal{M}$, the reference model $\widehat{\mathcal{W}}$, the hyperparameters $\gamma, \lambda, \omega$
**output** Updated coefficients $Y$, updated binary mask $T$
  **for** $l = 1, 2, ..., L$ **do**
    Initialize $\mathcal{W}_l \leftarrow \widehat{\mathcal{W}}_l, T_l \leftarrow 1$
    Reshape the weight tensor $\mathcal{W}_l \rightarrow W_l$
    Apply 2-D DCT to the original weight matrix $W_l \rightarrow Y_l$
  **end for**
  **for** $iter = 1, 2, 3, ..., iter\_max$ **do**
    Choose a minibatch of network input from $\mathcal{M}$
    **for** $l = 1, 2, ..., L$ **do**
      Reshape the input tensor $\mathcal{I}_l \rightarrow I_l$
      Apply 2-D DCT to the input matrix $I_l \rightarrow X_l$
      Forward propagation with $X_l$ and $Y_l$
    **end for**
    Loss calculation with $Y \otimes T$
    Backward propagation and generate $\bigtriangledown \mathcal{L}$
    **for** $l = 1, 2, ..., L$ **do**
      Compute the gradient of weight in spatial domain $\frac{\partial \mathcal{L}}{\partial W_l}$
      Apply 2-D DCT to the gradient $\frac{\partial \mathcal{L}}{\partial W_l} \rightarrow \frac{\partial \mathcal{L}}{\partial Y_l}$
      Update $Y_l$ using SGD method
      Update $T_l$ according to function $f(\cdot)$ and the current $Y_l$
    **end for**
    Update the learning rate
  **end for**

---