[Reviews · NeurIPS 2018]

Reviewer 1



In this submission, authors proposed a dynamic pruning scheme in the frequency domain to compress convolutional neural networks. Spatial-domain convolutional is converted into frequency-domain multiplication, in which weight pruning is performed to compress the model. A band adaptive pruning scheme is proposed to vary compression rates for different frequency bands. Experimental results show that the proposed method outperforms spatial-domain pruning methods. The manuscript is well organized and easy to follow. However, the empirical evaluation is not yet convincing (at least to me), and further experiments are required to improve the overall quality. a) ResNet on CIFAR-10. The reported baseline models’ accuracies are 91.23% for ResNet-20 and 92.81% for ResNet-110. This is considerably lower than results reported in other papers, e.g. 93.39% for ResNet-110 as in He et al.’s ECCV ’16 paper. b) How does the proposed method works on ResNet models on the ILSVRC-12 dataset? This can be more convincing than AlexNet for ILSVRC-12, since ResNet models are more commonly used recently. c) What is the actual speed up for the proposed method? There is only theoretical speed up reported in the manuscript. d) Is it possible to automatically determine the pruning ratio? For instance, use reinforcement learning as the external controller to choose the pruning ratio. e) For BA-FDNP, since different frequency bands have different compression rates, is it possible to apply structured pruning to improve the actual speed up? f) In Equation (12), the last term should be $1 / \xi$, instead of $\xi$.

Reviewer 2



[Response to rebuttal: Thanks to the authors for addressing my questions during rebuttal. My only major issue has been addressed and the same is true for my minor questions and issues, except for (5), which I do not consider crucial, particularly given that the authors only have one page for their response. Since most of my issues were regarding question that I had or minor detail that should be added to the paper, I have raised my confidence of reproducibility to 3. ] The paper introduces a novel method for parameter-pruning in convolutional neural networks that operates in the frequency domain. The latter is a natural domain to determine parameter-importance for convolutional filters – most filters of a trained neural network are smooth and thus have high energy (i.e. importance) in their low-frequency components, which means that high-frequency components can often be pruned without much loss in task-performance. An additional advantage of the method is that pruning is not performed as a single post-training step, but parameters can be pruned and re-introduced during training in a continuous fashion, which has been shown to be beneficial in previous pruning schemes. The method is evaluated on three different image classification tasks (with a separate network architecture each) and outperforms the methods it is compared against. The topic of neural network compression is timely and of high practical significance for deep learning applications. The paper is well written, although another pass to fix a few smaller glitches is recommended. The current state-of-the art and related work is nicely discussed in the introduction, though there are some recent methods that should be included and also compared against (see my comments below). In terms of originality, the method builds on two ideas which have been proposed recently: going to the frequency domain for neural network parameter pruning (references [14, 15] in the paper) and dynamic pruning and re-introduction of parameters during training (reference [18] in the paper). The paper combines both ideas in a novel and original way that makes a lot of sense and is demonstrated to work well empirically. The experimental section is satisfying, though not outstanding, except that comparison against some recent state-of-the-art methods is currently missing. I think this is a solid paper, which could be further improved to a great paper and therefore I recommend acceptance of the paper. My minor comments below are aimed at helping the authors to further improve the quality of the paper. Major: While the Introduction generally gives a good overview over state-of-the-art approaches in network compression, there are a few methods that should be mentioned and if applicable the methods should also be compared against in the experimental section as these are spatial-domain pruning methods that perform on-par with the proposed method (luckily, most of them report results for LeNet-5 on MNIST, so there is no strict need to run additional experiments). Most of the methods use the idea of inducing sparsity during training via learning a posterior-distribution over weights under a sparsity-inducing prior. This paradigm has been remarkably successful and at least some of the methods have the advantage that they prune whole neurons/feature-maps, which leads to speed-ups without requiring special hardware. Please briefly discuss and compare against the following methods (I do not expect the authors to run additional experiments, all papers report results on LeNet-5, additional results and comparisons are a bonus of course): Weight pruning: [1, 2] Neuron/filter pruning: [3, 4, 5] [1] Ullrich, Meeds, Welling, Soft weight-sharing for neural network compression, ICLR 2017 [2] Mochanov, Ashukha, Verov, Variational Dropout sparsifies neural networks, ICML 2017 [3] Louizos, Ullrich, Welling, Bayesian compression for deep learning, NIPS 2017 [4] Neklyudov, Molchanov, Ashukha, Vetrov, Structured Bayesian pruning via log-normal multiplicative noise, NIPS 2017 [5] Louizos, Welling, Kingma, Learning sparse neural networks through L0 regularization, ICLR 2018 Minor: (1) Sec. 2.3: In FDNP, the importance of a parameter is given by the frequency-coefficient (the energy of the frequency component). I find it natural to prune frequency components with small coefficients. However, in BA-FDNP the paper argues that frequency components have different magnitudes and thus proposes to bin frequency components and apply different pruning thresholds to each bin. Empirically this seems to work even better than FDNP, but I struggle a bit with the argument because this seems to go against the original argument that the frequency coefficient determines parameter importance. Do I misunderstand something or are you essentially saying with BA-FDNP that “you want to keep a certain number of parameters from each frequency band, even though potentially all the coefficients of that band have a very small magnitude compared to other bands”? (2) It would be nice to show a histogram of typical frequency components of a trained network and a pruned network. Perhaps this would also clarify the questions of my previous comment (1). Additionally it would also be interesting to see a few examples of pruned filters which have been transformed back into the spatial domain and compare them against the original, unpruned version. (3) Question regarding the training procedure: what is the loss function in Eq. (4) used in your experiments, standard cross-entropy loss? Why does it only involve a single layer though? In the pseudo-code (Appendix 2) it looks like you compute the loss for each layer and then perform a gradient-update step on the spatial-domain weights for each layer. I would have expected that you need to do the full forward pass through all layers first and then compute the loss and back-propagate through all layers accordingly. Did I miss something here? (4) I greatly appreciate that the authors are clear about the reported speed-ups being of a theoretical nature. Please also indicate in the abstract (line 11) that the speed-up is a theoretical one. (5) Line 143 – 146 and Eq(8): You mention that you evaluate several methods for setting the compression rate of the individual frequency bands. Could you please add some more details on alternatives that you have tried or propose to try (perhaps in the appendix?) and give some arguments on why you finally chose the function in Eq(8)? (6) Line 27: I think the reference you have here ([5] in the paper) only has binary weights but not activations, the one you are referring to is from the same main author, but the title is: Binarized Neural Networks: Training deep neural networks with weights and activations constrained to +1 or -1. (7) Reproducibility of experimental results: for all experiments please report: which gammas did you use in the experiments where you set them manually (line 197)? Do you use a learning-rate schedule, if so which one? What initialization method did you use, which optimizer with which parameters? Did you do any pre-processing / augmentation of the data? (8) Line 227: You mention that you prune “the other fully-connected layer in spatial domain” – with which method? Reference [18] from the paper? (9) Line 246: Same question as before: which method do you use for “spatial-domain dynamic pruning”? (10) Can you comment on whether a pre-trained network is strictly necessary for your method or whether there is also the potential to train from scratch?

Reviewer 3



The work reported in this paper is focused on reducing the computational cost of convolutional neural networks in image processing. The major strength of this paper is the conversion of the problem into the frequency domain, as outlined in Equations 1-4, which is also appropriately illustrated in Figure 1. The pruning in frequency domain and band adaptive pruning is also a nice extension. The major weakness of this work is that most of the gain in the computational cost is available only on hardware where the multiplication with zero is not an operation (nop) and can be smartly skipped. However, most CPU and GPU hardware are not currently geared to take advantage of the savings associated with multiplication with zero. The empirical results also do not comment on practical savings in computational cost and that is a disappointment.